# Quantification of the Starling Population, Estimation and Mapping of the Damage to Olive Crops in the Apulia Region

**DOI:** 10.3390/ani11041119

**Published:** 2021-04-14

**Authors:** Francesco Bozzo, Simona Tarricone, Alessandro Petrontino, Prospero Cagnetta, Giacomo Maringelli, Giuseppe La Gioia, Vincenzo Fucilli, Marco Ragni

**Affiliations:** 1Department of Agricultural and Environmental Science, University of Bari Aldo Moro, 70125 Bari, Italy; francesco.bozzo@uniba.it (F.B.); simona.tarricone@uniba.it (S.T.); prospero.cagnetta@uniba.it (P.C.); giacomomaringelli73@gmail.com (G.M.); vincenzo.fucilli@uniba.it (V.F.); marco.ragni@uniba.it (M.R.); 2Or.Me. via Saponaro 7, 73100 Lecce, Italy; ormepuglia@gmail.com

**Keywords:** starling monitoring, olive crops, agriculture, damages, spatial analysis

## Abstract

**Simple Summary:**

Near Brindisi (Apulia, S-E Italy), a substantial amount of cash compensation claims for damages reported by local farmers and attributed to starlings (*Sturnus vulgaris*) has been registered. This led us to conduct this study in order to quantify the starling population wintering in the Apulia region and to assess the potential damage to crop production. Our analysis was conducted over three years, to determine species abundance, their movements, and their preferred areas and crops. The study showed a loss, in terms of gross profitable production, of around 550,000 euros per year, concentrated mainly in a few limited areas. Results on species behaviour, damage quantification, and mapping are useful elements aimed to activate trade-off measures to preserve production and protection objectives, and to allow policymakers to address enforcement interventions and to establish parameters for financial compensation.

**Abstract:**

The presence of wildlife in areas with a high concentration of farming activities can create a conflict between conservation objectives and productive purposes. Near Brindisi (Apulia, S-E Italy), a substantial amount of cash compensation claims for damages reported by local farmers and attributed to starlings (*Sturnus vulgaris*) has been registered. The aim of this study was to quantify the starling population wintering in the Apulia region, in order to assess the potential damage to crop production caused by this species. Our analysis was conducted over three years and included three main activities: a study of starling abundance and movements, the identification of areas and crops affected by damages, and a determination of the damage to the agricultural system in terms of quantity and concentration (heatmap). The study showed a loss of expected production that was coherent with the eating capacity of starlings wintering in the region. This means a loss, in terms of gross profitable production, of around 550,000 euros concentrated in a few narrow areas close to the roosts. Results on species behavior, damage quantification, and mapping are useful elements aimed to activate trade-off measures to preserve production and protection objectives, and to allow policymakers to address enforcement interventions and to establish parameters for financial compensation.

## 1. Introduction

The Apulia region is one of the most important resting areas in Italy for many migratory species, including starling (*Sturnus vulgaris*) [1]. The climate of this region is typically Mediterranean, characterized by abundant precipitation in the autumn and winter seasons (max of 850 mm/year) and dry summers. The landscape is mainly composed of olive orchards (nearly 75%), and natural vegetation is represented by Mediterranean maquis. Agriculture, in particular olives, olive oil production, and viticulture, is the most important economic sector of the area, but in the last few years many olive producers have denounced the damage caused by starlings.

### 1.1. Starling Population in the Apulia Region

The starling (Sturnus vulgaris) is a gregarious medium-sized passerine with a short tail and long bill, and it moves in flocks of various sizes [2]. In Apulia, the first breeding starlings were observed in Bari in approximately 1965 and more recently have extended to the entire region; today, it is considered to be a regular migratory, wintering, and resident breeding species [3]. Populations wintering in Apulia come from a wide area stretching from Central Europe to Russia, and the specimens in transit reach Tunisia [4]. In Apulia, during the winter, nesting and migratory groups coexist, and the starling density reaches the highest values between early October and late February, a period coinciding with the maturation and harvest of olives.

The large starling population of wintering in Apulia cause damage to agricultural systems such as to horticultural and cereal crops, and to fruit and grapevine production. The most recurrent attacks reported from farmers consist essentially towards olive growing and are related to the withdrawal of fruits from trees and the ground. Considering the period of the greatest presence of starlings in Apulia (November–February), olives are the most commonly affected culture, in particular where ground and late harvesting is practiced. The damage caused to harvest beds, horticultural crops, arable lands, and feed for livestock, although noticed, was not significantly appreciable.

The strong relationship between the density of the birds and the extent of the damage in Apulia, based on farmer reports collected in the past, is due substantially to cultivation practices, in particular to the harvesting period. The harvesting is scalar following the ripening of fruits and is often performed by picking up olives from the ground. The starling, on the other hand, at the beginning of the olive-growing season (October), is not yet numerically relevant, and its diet is predominantly insectivorous. Subsequently (November–December), with lower temperatures, there is a considerable increase in the starling population, until it reaches the maximum value detected of a few millions, and its interest in olives increases due to olive ripening and to the decrease in the number of insects (Figure 1). This is the period with higher fruit withdrawals. Therefore, the damage detection was concentrated in areas where the scalar and late harvesting of olives is practiced.

The main damages were recorded near the coast on a surface of about 100,000 ha, 40,000 of which was occupied by olive groves. These areas are located close to Brindisi roosts, on a territory of about 70 km in length and 10 km in depth.

### 1.2. Quantification and Distribution of Damages Caused by Starlings

The estimate of damage resulting from starlings is extremely complex because of the uncertainty of the temporal and spatial distribution of attacks. Several European studies [5,6], based on expert assessment, have estimated the damage on crops (from 0.25–3.8%) around the starling roost. A research program of the Liguria Region estimated a presence of 500,000 starling wintering for 80 days and a daily withdrawal of 10 olives per individual [7]. Seasonal total consumption of olives amounted to 720 tons equivalent to an approximately 5% loss of production [8]. The same calculation applied to real data recorded in 1999 leading to a damage assessment of 0.52% [7].

There is evidence from the literature of a lack of spatial homogeneity, suggesting a need for a spatial analysis to circumscribe the area in which starling feeding may have provoked the greatest effects. The application developments of spatial interpolation techniques cover different technical and scientific fields. In hydrological studies [9], the geomorphological information forms the basis for the construction of digital models of the ground or the estimate of outflow water. In ecology studies [10,11,12], the relationships between the presence of wild species and the environmental parameters measured in scattered points are particularly useful for creating species spreading maps, for estimating the population, and for identifying trophic areas; in meteorology, they are useful for the construction of climate maps based on data from different countries [13].

How points in space are interpolated can be different and follow different rules that are generally formalized through specific mathematical models [14,15,16]. Moreover, studies applying spatial analysis generally face a choice between deterministic and geostatistical methods, where deterministic interpolation methods predict the values at points in space, based on deterministic functions (variables can assume one and only one value at a given time). Geostatistical interpolation methods, instead, use the statistical properties of the measured points, incorporating the concept of randomness. The interpolated surface is therefore conceptualized as one of the possible surfaces that can be observed, all however consistent with the measured data [16]. Uncertainty related to starling behavior imposes a geostatistical interpolation method in order to take into account the randomness of seasonal weather trends, wind, microclimatic conditions, and the availability and attractiveness of food [17]. In particular, some studies have focused on kriging, one of the most commonly used geostatistical methods for estimating and mapping species distribution. Karanth et al. [18] and Chen et al. [19] show how this method can be used for the evaluation of damages caused by wild animals to crops.

## 2. Methodology

Our analysis was divided into two sections that need specific knowledge: (i) a study of starling abundance and movement around their roosts and the identification of areas and crops affected by damage in Apulia (2.1); (ii) a determination of the total damage to the agricultural system and the interpolation of data collected to obtain a heatmap of the damage (2.2).

The indicated activities have been repeated for three different wintering periods of the species: 2016–2017, 2017–2018, and 2018–2019.

### 2.1. Estimate of Starling Abundance, a Study of Their Movements around Roosts, and the Identification of Areas and Crops Affected by Damage in Apulia

The localization in Apulia of the main nocturnal aggregation sites of starlings was carried out using two methods. The first, indirect, made use of a capillary network of collaborators (colleagues, hunters, birdwatchers, farmers, supervisors of natural and/or protected areas, etc.), who were asked for systematic information throughout the investigation period about the presence/absence of this species in certain areas of the region. The second method consisted in the direct verification of starling presence as confirmation of the information received from collaborators. Our detection was performed monthly from October to February on each roost to obtain a good indication of the numerical consistency of the wintering population in the entire region.

Since the starling preferentially roosts inside large reed beds (*Phragmithes australis*) or in the thick foliage of trees, often in the woods, it is not possible to count them once they have landed. To count these birds, we used the estimation method suggested by Bibby et al. [20], the method applied during the last stages of approaching flocks to roost and before any navigation flights within the same site. This technique involves estimating or counting a “group” of birds inside the flock (e.g., 5, 10, 20, 50, 100, 500, 1000, or more birds), depending on the size of the flock and the size of the birds. The “group” is then used as a reference unit to quantify the number of other birds that make up the flock. During the census, we paid particular attention to when the flock showed different densities.

In roosts composed of a large number of birds that came from different directions and occupying a large portion of suitable territory, the number of birds was also counted when, occasionally, they flew up at the same time, often due to attacks by predators.

For the numerical estimation of roosts, for each site, 1–6 simultaneous surveyors were distributed up to and beyond 3 predetermined points, situated from 2 h before sunset and, therefore, in good time to observe all animals going back to the roost. Occasionally the census was also carried out at sunrise when the birds left the roost, heading quickly toward the feeding areas.

During this activity, we carefully studied the terrain, recorded the main directions of origin or disappearance of the animals that made up the roost, and identified the best observation sites for subsequent observations, to make more accurate numerical estimates and to understand the directions of travel to and from the roost, useful for research in foraging areas.

During the daytime, numerous inspections were carried out to identify the trophic areas. The inspections, performed with the aid of suitable optical instruments (binoculars and telescopes) and video-photographic equipment, were often started in areas close to those of the main roosts; subsequently, the researchers went in several directions, moving away to investigate a larger portion of the territory [21].

To give an overview of the areas frequented by starlings in this vast region, they were grouped and therefore represented in greeds of a 10 km Universal Transverse of Mercator (UTM) pattern that overlaps Apulia. During these trips, frequent interviews were carried out with farmers or residents of the visited areas to acquire indirect information. Special questionnaires were also distributed to trade associations and other stakeholders in the agricultural field.

### 2.2. Determination of the Total Damage to the Agricultural System and the Interpolation of Data Collected to Obtain a Heatmap of the Damage

Based on the data collected through the interviews and questionnaires done during the activity described in Section 2.1, we identified areas more frequented by starlings. In these areas, we collected data on farm location, crop extension, current year of expected olive production, and the price sales of the olives harvested (€/kg). The expected production was estimated by researchers referring for each farm to the mean production of the last 5 years. Afterwards, the damage to single farms was obtained as a percentage loss:
$$D=\frac{\left(Ye-Yf\right)}{Ye}*100$$
where

***D*** is the damage expressed as a percentage;

***Ye*** is the expected production;

***Yf*** is the final production.

Finally, a heatmap showing the concentration of the trophic activity of the starlings according to the estimate of the average damage per hectare was created. This step started with the georeferencing of the investigated farms. The set georeferenced point to be used for the interpolation imposed a series of unavoidable approximations related mainly to the nature of the data detected. The data collected on a farm’s references were attributed to each corresponding map sheet. Subsequently, the damage value detected for the single map sheet was concentrated at a single point coinciding with the centroid of the same sheet.

Given the impossibility of discriminating different levels of damage for each sheet, the damage value was attributed equally on all the sheets with which the farm was composed. If a map sheet contained more than one farm or portions of farms, a unique value was assigned, obtained by averaging the individual percentages of damage recorded on the same sheet by each of the farms involved. An interpolation of collected data was done to obtain a continuous surface able to show the gradients of damage produced on each olive field in the survey area.

The method used for creating the heatmap is kriging. This is a geostatistical regression method that allows the interpolation of numeric data in space, minimizing the average quadratic error [22]. This type of interpolation incorporates spatial autocorrelation, that is, the statistical relationships between the measured points. Kriging also assumes that the distance or direction between the sample points reflects a spatial correlation that can be used to explain variations in the prediction surface. The unknown value in each point is calculated with a weighted average of the known values, according to the following formula:$$Ž\left(S_{0}\right)=\overset{N}{\underset{i=1}{\sum }}\lambda_{i}Z\left(S_{i}\right)$$
where
-Ž (*S*_0_) is the predicted value;-*S*_0_ represents the position of prediction;-*N* is the number of measured points.-*λ_i_* is the unknown weight for the measured value in the ith point;-*Z* (*S_i_*) is the measured value of the ith point;

This geostatistical method provides a prediction surface and a map of reliability that takes into account the values found and the mutual position of the detected points [15]. The accuracy of predicting surface was used to discriminate or exclude those areas where the degree of approximation would be too high. Finally, an overlay operation with a land cover map (Corine Land Cover map of Apulia, 2011) [23] allows the production of a forecast map [24,25] exclusively of the olive-growing areas (Figure 2).

## 3. Results

### 3.1. Quantification of Starling Population and Movements around the Roosts

During the study period, seven roosts were identified with more than 20,000 starlings counted in at least one monthly visit: Lago Salso and Valle San Floriano in the province of Foggia (FG), Ariscianne in Barletta-Andria-Trani province (BAT), Torre Canne, Torre Guaceto, and Fiume Grande in the province of Brindisi (BR), and Bacini di Ugento in the province of Lecce (LE). The swamp of the Baia Verde in the province of Lecce, the mouth of the Fiume Lato, and that of the Torrente Galeso, in the province of Taranto (TA), have shown medium importance, with some specimens numbering between 1000 and 20,000. Lago di Lesina (FG), the mouth of the Fiume Patemisco (TA), the Palude del Conte (LE/TA), Le Cesine, and the Lago Alimini (LE) have local importance with less than 1000 specimens (Figure 3).

All the roosts identified were located in wetlands with reeds, even if the presence of small roosts in the pine forest of the high Taranto coast, or that of the Island of San Pietro in Taranto’s Mar Grande, once very consistent, is not excluded.

The Lago Salso–Valle San Floriano (FG) and Torre Guaceto–Fiume Grande–Torre Canne (BR) groups can be considered as two locations of a single territorial grouping of starlings: within the same wintering season, it was possible to observe how this species occupied all the localities of the same group with variable percentages (Table 1).

The roost located in Ariscianne, although presenting high values, seems to be used for a shorter period than the others and not in all the years, while the Bacini di Ugento roost was occupied by a greater number of specimens only rarely and for a short period.

Roosts with high regional importance are home to most of the wintering specimens in the region (Table 2) estimated at a few million, with a maximum generally in January.

### 3.2. The Existence of Damages

The highest density of starlings was identified in the areas between Torre Guaceto and Torre Canne and between Lago Salso and Valle di San Floriano. Therefore, the survey focused on these areas extending to a radius of about 70 km (Figure 4).

In the first year of surveys, a loss of 1,375,000 kg of olives was estimated exclusively in the area adjacent to the Torre Guaceto roost. Considering an average price in the reference area, the value of the loss was estimated at approximately 550,000 euros. In the second year, researchers revealed on these farms a damage of 800,000 kg of olives, which, based on the declared sales prices, corresponds in value to a loss of approximately 447,000 euros. During the third year of study, 1,285,000 kg of olives were subtracted from the farms surveyed, which in terms of value corresponds to a loss of about 535,000 euros. Though the spread of the olive-growing areas is almost homogeneous, especially on the coastal plain between the provinces of Bari and Brindisi, the percentages of damage suffered by farmers as well as the expected production yields are spread in a very different way. The study of olive farms showed an average loss of 12.6% of the expected production over the entire investigated area with a very low uniformity of distribution.

The heatmap allows for an estimation of how the average loss is distributed over the regional territory. The analyses carried out are based on the assignment of the damage percentage on the map sheets belonging to each farm and on the interpolation of the values. As regards the propaedeutic geo-referencing operations for data interpolation, the first year was assigned damage values on 119 map sheets. In the second year, there were 354; in the third year, 364 map sheets were assigned.

The extent of territorial distribution of the detected damages was observed by mapping the kriging interpolation of map sheet damage values. The analysis produced a second map showing the variance of the data collected over the region. This represents the reliability of the heatmap. Variance values were split into three classes (low, medium, and high) obtained from an “intelligent quantiles” classification. The result of the interpolation was therefore limited only to the olive groves falling into areas with low variance (Figure 5).

The map in Figure 5 shows the evolution of damage during the three years of the study, revealing the hot spot where damage exceeded 30%. In the first year, a particular concentration around Torre Guaceto roost was recorded: the average damage value is 35%, while the maximum and minimum values are 60% and 23%, respectively. Concerning the area of damage prediction deriving from the extraction of the olive-growing areas, only a slight increase is recorded.

In terms of the second year, the map shows the first nucleus with high damage (62%), which is located within an approximate radius of 10 km extended from the municipality of Ostuni towards Fasano in the northeast and towards Francavilla Fontana in the south direction. The second nucleus, with milder damage (45%), is instead located between Corato, Ruvo, and Palo del Colle.

In the third year, the first nucleus with high damage was located at a distance of 10 km from the dormitory of Torre Guaceto. This area extends to the border of the municipalities of Ostuni and Carovigno and constitutes the damaged area closest to the dormitory. The second area is instead located about 45 km from the roost of Torre Guaceto and less than 10 km from the roost of Torre Canne. The harvest of olives found in the farms of this area also stands at 30% and constitutes the largest damaged area, almost entirely affecting the municipalities of Fasano and Monopoli. The third area with damage greater than 30% is ambiguous in terms of the origin of the starting roost. It consists of the territory between Palo del Colle and Toritto, presumably attacked by starlings starting from Lago Salso or Valle San Floriano.

## 4. Discussion

All the seven most important roosts (Figure 3) showed highly variable importance during the three years of monitoring or in the months of the same year. The variation is much greater than the trend attributable to the migratory phenology of the species: in the same wintering season, this underwent drastic numerical variations, often for no apparent reason. In fact, roosts of several million specimens were found to be completely disappearing from one location and forming in another. For example, in the winter of 2016–2017, there was a significantly lower wintering starling population in Apulia (3 million vs. 9 and 10 million) than in the following two years. This may have happened because January was characterized by abundant snowfall, which led to a fragmentation of the roosts and a shift towards the south in areas without snow, which prevents the finding of food on the ground and a greater difficulty for the census.

The main movements of specimens between roosts were observed between Torre Guaceto, Fiume Grande, and Torre Canne, in the Brindisi area, and Lago Salso and Valle San Floriano, in the Foggia area. High turnover rates at roosts have also been shown in other studies [26,27]. Daily, starlings can travel long distances from the roost, up to 100 km, with cruising speeds of up to 80 km/h [21,28]. During this flight, they tend to disperse into smaller groups as they move away from the roost. The direction of movement probably changes depending on trophic availability during the winter. The greatest distances have been recorded for the roosts of the Foggia area: these starlings are distributed over a wide front that goes from the olive groves of the Gargano to the Monti Dauni and the olive groves of the BAT attending the intensive arable land of the Tavoliere, mainly during cereal sowing.

The roosts of Brindisi are immersed in a more favorable environment and, therefore, find trophic areas at shorter distances and, consequently, have the most numerous feeding groups. When starlings arrive to the roost, the majority arrive from the west and northwest directions, where olive groves are concentrated. Occasionally, they arrived from the southeast and on those occasions our technicians found them in the olive groves near Lecce during the day. Transfers have been spotted mostly in areas with a mosaic of olive groves, orchards, vineyards, and arable crops (vegetables, cereals, and fallow), sometimes even near population centers or corporate agricultural settlements. Finally, there have been observations in extensive and continuous olive groves.

Starlings are predominantly insectivores but also eat vegetable sources (olives, Bay Laurel, etc.), fleshy fruits (prickly pear and dried wine grapes), and seeds (cereal taken during sowing) [25,29,30,31]. Nevertheless, the presence of starlings in the cultures mentioned above does not imply that such specimens feed exclusively on their production: in fact, the presence of a large number of starlings in olive groves and vineyards already subject to a complete harvest has also been ascertained. This event can be explained by the fact that these birds seek nourishment in the soil, easily accessible because of poor or no herbaceous vegetation. It was also found that the starlings actively seek their prey even in the cracks of the bark of the vine. Regarding the use of olives as a source of food, it was verified that, if the olives are small, they can be swallowed whole directly on the tree. The larger olive varieties, like other similarly sized drupes [32,33], are transported in the bill in areas with good visibility, such as those with power cables or open terrain with sparse vegetation, where smaller portions are taken from the olives for easy ingestion.

Several farmers have complained about damages to cultivation plots due both to the removal of olives from the ground and to the displacement of soil for insect search. Since it is difficult to assign this kind of damage to starlings, our estimation was based only on the calculation of the loss of product according to the difference between the expected production (on the basis of fruits on trees) and the product obtained at the end of each harvest.

The olive production loss due to starling attacks was estimated considering two issues: (i) production estimated by a university researcher referred to a time frame of the last five years of full production; (ii) production obtained in 2016 was part of a discharge period. The difference between final production (Yf) and expected production (Ye) generates an amount attributable to the combined effect of the following factors: production alternation, possible differences between the five years of reference and 2016 in terms of climate variability, variations in agricultural practices, and different incidences of plant diseases. This amount was subtracted from by the activity of starlings. In other words, we came to a difference that is a gross value of the “starling phenomenon”, and a series of exogenous causes was considered insignificant for the study.

The damage analysis represents a further element to support the behavioral evidence that emerged during the observation of the species [34,35]. The study showed that, in the three years, a loss of expected production over the entire area investigated was highly coherent with the eating capacity of starlings wintering in the areas investigated [7]. This means that, during the olive-growing season, about a million starlings daily presumably ate 5–6 olives weighing three grams each (equal to about 1,400,000 kg), thus generating a loss, in terms of gross profitable production, of around 550,000 euros.

The result obtained appears to be in line with what has been observed in Liguria [7], where a group of 500,000 specimens collected approximately 720,000 kg of olives. The results obtained through spatial interpolation (35% of the damage) were higher than the average obtained with the estimation of the damage on a farm basis. This suggests that the delimitation of a forecast map, due to the selection of variance, can effectively achieve the objective of identifying and circumscribing a trophic route, which is territorially smaller, but higher in terms of damage intensity.

The area where the greatest damage was recorded is located at a significant distance from the roosts. For example, from Torre Guaceto, the sub-area of Ostuni is about 22 km away; the area between Fasano and Monopoli is about 40 km away, while the area of Monopoli, the most affected in terms of extent of damage, is 50 km away. Other authors have shown similar distances for the foraging flight of starlings [36]. While in the two southern roosts and adjacent areas the strong relationship between the density of birds and damage is substantially attributable to the cultivation technique (late harvesting), in Lago Salso, despite a huge population, farmers do not complain of any significant damage. Results from the questionnaire show that no particular preferences were observed, except for ripe fruits, while several farmers have also complained of damage to cultivation areas due to the removal of olives from the ground. The areas most affected are also those in which there is late maturation, coinciding with the maximum population of the roost in December, and those in which the presence of traditional olive groves, as well as secular groves, in which drupes remain on the plant until fully ripe or are collected from the ground, is more frequent. These significant aspects will certainly be investigated in more depth and be taken into account in future monitoring operations.

## 5. Conclusions

Apulia is confirmed as one of the Italian regions hosting the largest number of wintering starlings (from three to ten million), which are currently divided mainly into two groups that form their roost in the Lago Salso or Valle San Floriano, in the Foggia province, and Torre Guaceto, Fiume Grande, or Torre Canne, in the Brindisi province.

At a regional scale, a loss varying from 447,000 up to 550,000 euros was recorded. The investigation conducted in the feeding area of starlings belonging to the roost of Brindisi indicated more harmful activity for olive growing in the territories of Monopoli, Fasano, and Ostuni (damage over 30%). Moreover, according to spatial analysis, the starlings prefer to explore the routes to the northwest of the roost of reference, also covering considerable distances.

The results of the study lead to some useful findings, both in scientific and management terms. To improve the monitoring and estimation of the damage produced on olive groves in future years, the principles underlying the trophic behavior resulting from this study should be considered. Results on species behavior, damage quantification, and mapping are important parameters to activate equal and adequate trade-off measures between the economy and the environment. This will allow policymakers to address enforcement interventions and to establish factors for financial compensation to preserve production and protection objectives in a defined geographic area.

Finally, it would be desirable to raise awareness among olive farmers, particularly those whose lands fall within the trophic route, of the numerous aspects that have emerged and to encourage the adoption of cultivation guidelines that also take into account the presence of starlings in the territory. This aspect could be pursued to anticipate the movements of the animal based on climatic trends and production forecasts, encouraging also harvesting techniques and times aimed at limiting as much as possible the starlings’ withdrawals.

## Figures and Tables

**Figure 1 animals-11-01119-f001:**
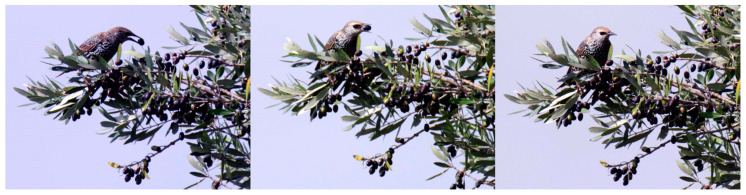
A starling eats an olive on a tree (ph. La Gioia, G.).

**Figure 2 animals-11-01119-f002:**
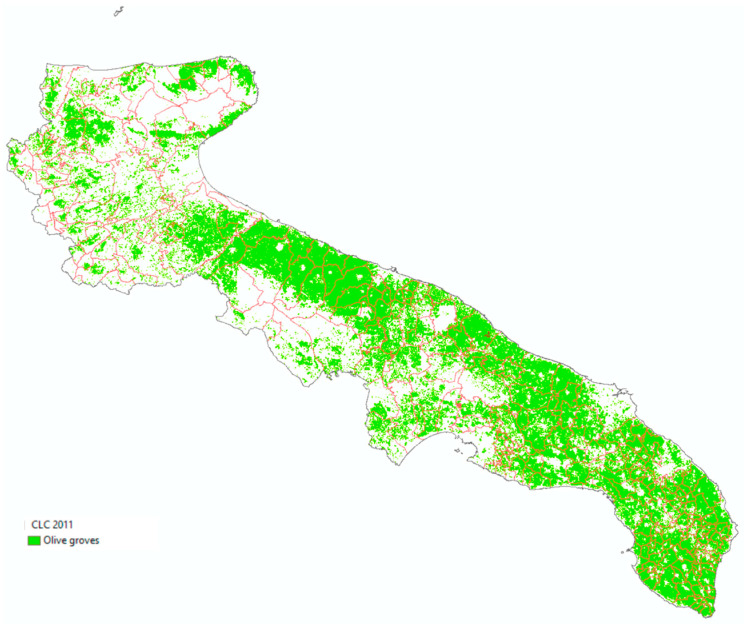
Olive groves in the Apulia region (Corine land cover 2011).

**Figure 3 animals-11-01119-f003:**
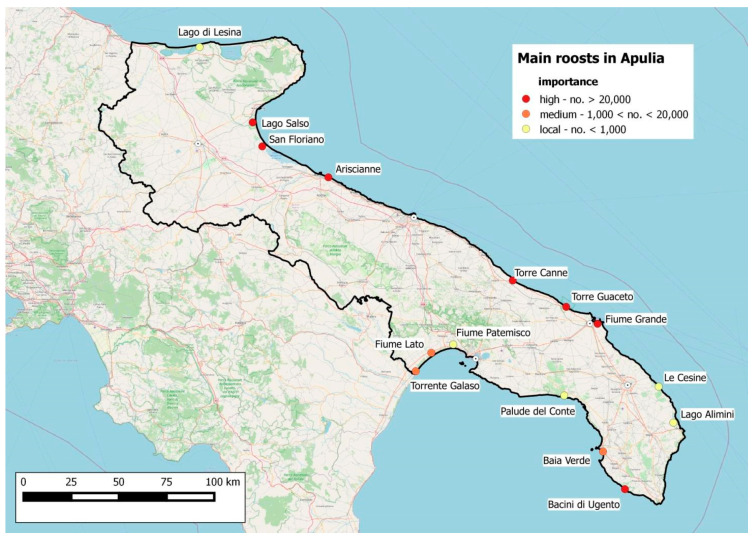
Identification of roosts and inspection areas within 70 km radius.

**Figure 4 animals-11-01119-f004:**
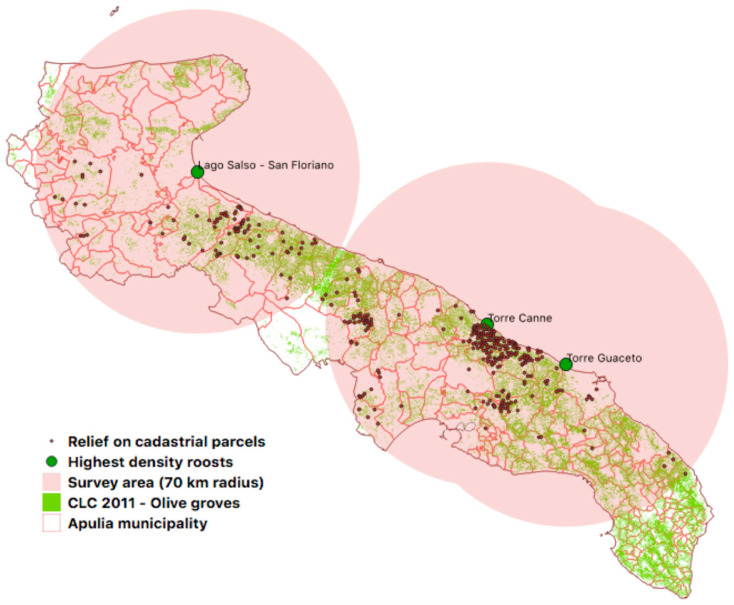
Identification of roosts and inspection areas within a 70 km radius.

**Figure 5 animals-11-01119-f005:**
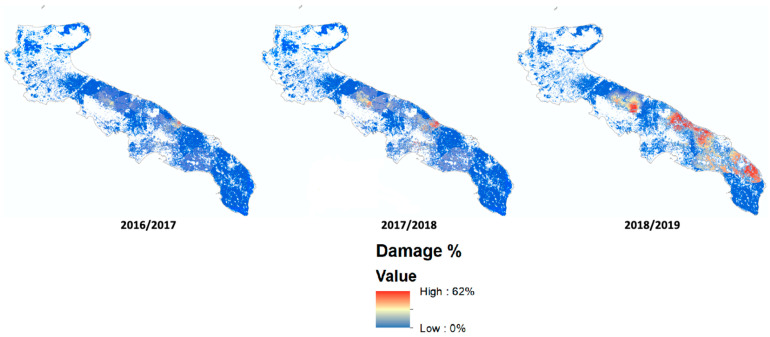
Distribution of damaged areas resulting from kriging interpolation. Areas over 30% damage are identified in red.

**Table 1 animals-11-01119-t001:** Monthly estimate of the population size of wintering starling in Apulia and the main roosts in the three winters of the study.

Winter	Main Roosts ^1^	October	November	December	January	February
2016/2017	FG BR	45,000 20,000	500,000 470,000	1,400,000 1,500,000	1,400,000 1,600,000	1,500,000 1,000,000
2017/2018	FG BR	50,000 25,000	1,000,000 1,300,000	1,200,000 1,300,000	2,000,000 7,500,000	1,500,000 2,000,000
2018/2019	FG BR	50,000 30,000	1,700,000 1,300,000	6,000,000 2,500,000	7,000,000 3,000,000	1,500,000 1,000,000

^1^ Main Roosts: FG—Lago Salso—Valle San Floriano roosts; BR—Torre Guaceto—Fiume Grande—Torre Canne roosts.

**Table 2 animals-11-01119-t002:** An estimate of the population size of wintering starling in Apulia and the main roosts in the three winters of the study.

Winter	Apulia Population Maximum Size	Main Roosts ^1^	Roost Maximum Size	SE ^2^
2016/2017	~3 million	FG BR	1,400,000 1,600,000	±15,000 ±20,000
2017/2018	~9 million	FG BR	2,000,000 7,500,000	±20,000 ±30,000
2018/2019	~10 million	FG BR	7,000,000 3,000,000	±25,000 ±20,000

^1^ Main Roosts: FG—Lago Salso—Valle San Floriano roosts; BR—Torre Guaceto—Fiume Grande—Torre Canne roosts; ^2^ SE: Standard error.

## Data Availability

Data are not available for public consultation.

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
