# Peer review of "Quantification of the Starling Population, Estimation and Mapping of the Damage to Olive Crops in the Apulia Region"

_animals, 2021, doi:10.3390/ani11041119_

Round 1
Reviewer 1 Report
Thank You for an opportunity to review this paper, the topic is very interesting but I see several shortcomings of this paper.
- Information about starling biology is not important for whole paper understanding and I recommend to cancel it, the whole manuscript is generally to long.
- This section is also to long and partially difficult to understand. I really doubt that one starling can eat 10 olives per day, cited literature (7) is not relevant, it’s difficult to assess quality of this data, as I understand it is a book or some technical report ?
- In this section should be presented a map, moreover I don’t see any cited literature in this section. Is it the Authors’ own unpublished data or part of the results?
- This section I also to long, Authors presented here some methodological information mixed with theoretical background of the research as a result the text is difficult to understand and in a big part redundant.
Methodology - In this section only one scientific paper is cited. The methods are mixed with the some part of the results (lines 225-226) or divagations.
In my opinion Authors generally apply not adequate methodology: the study area is too vast and it’s impossible to do detailed survey in this scale. I suggest to limit the study area to one rusting site, apply the telemetry technique to assess the main foraging areas. In order to evaluate damages done by animals Authors should arrange the field experiment by protecting part of foraging site by nets and compare olive crop with not protected part of olive orchard. I think it’s the only reasonably method. Authors also should take under consideration that not only starling eat olives but also other animals.
In table 1 standard errors should be presented
Lines 357-358 Is it the Authors’ data? Because I don’t see any cited literature.
Lines 364-365 How did Authors assess which roosting sites are more favorable?
Figure 5 is unreadable
Discussion section is very poor because only one paper is cited, the rest these are divagations not supported by scientific high quality literature or Authors results presented above. Generally literature list is highly limited, only 25 positions.
Reviewer 2 Report
This is interesting scientific problem and undoubtedly is worth publishing but article should be completely rebuilt and shortened. The introduction and the discussion require need to be firmly placed the background and to focus more on the effect of feeding animals (birds) on crop damage in general. The results should be shortened and given more precisely. Compared to the other chapters, the discussion is relatively short and does not meet the requirements this type of paper. The references mainly include works on starlings, there are no other articles on similar problems.
The paper is very chaotic and the particular sections contain text that should be present in other part of the manuscript. For example: chapter 1.1 in the introduction is species description and should be presented in the methods, chapter 1.3 in the introduction is a part of the study area, chapter 1.4 in the introduction discuss the methodical assumptions, chapter 3.2 in the results is mostly part of the discussion.
Language and style are need a corrections. Some sentences are too long, which makes them difficult to understand e.g. 317-325, 331-337, 383-387.
Not all figures are cited in the text. After citation of figure 3, the figure 6 appears, which is not included. Figure 3 needs to be edited because it is not made to generally accepted standards.
Reviewer 3 Report
Review: Quantification of Starling population, estimation and mapping of the damage to olive crops in Apulia Region
By Francesco Bozzo, Simona Tarricone , Alessandro Petrontino, Prospero Cagnetta , Giacomo Maringelli , Giuseppe La Gioia , Vincenzo Fucilli , Marco Ragni
General comments:
- I think the presented idea is fine but also that the manuscript needs quite some work to be ready for publication
- It was very difficult for me to follow the story through the paper from introduction to conclusion
- While the Introduction and Methods sections are easy to read, I lost track from the Results section and onwards. The results section is very long and could be improved by a better structure and by being shortened
- The Discussion section is on the other hand surprisingly short and I am not sure that it reflects the previous sections
- The Conclusions are very long and cannot bring new parts into play
- There are no references in the text to fig 4 or fig 5. There is a reference to a fig 6 that is not included. The use of fig. and figure is not consistent.
- I suggest to include the information in fig 1 and fig 2 in fig 4
- The manuscript would benefit from giving more information on why this has become a bigger problem than it was years/decades ago and also to describe why the results are important and how they should be used in management.
- Is the winter population increasing? Is the breeding population from where the starlings originate increasing; see https://pecbms.info/trends-of-common-birds-in-europe-2019-update/
- It would also benefit from a language proof by a native English speaker
Minor comments:
Line 15-17 ‘550 thousand euros’ – annually or for all three years?
Line 38 ‘Most important areas’ compared to what?/in the world?/in Italy?
Line 39 I suggest to write …including the Common Starling Sturnus vulgaris, hereafter Starling
Line 40 ‘precipitations’ – you will need an adjective to describe the amount
Line 46-51 Irrelevant, delete
Line 59-61 Since the damage is caused outside the breeding season, the important part must be the huge flocks of migrants from northern breeding sites
Line 66-67 I don’t understand the last half of the sentence - after ‘…to record…’
Line 70 Do you mean ‘…in each of three different…?
Line 72 In this section, if possible mention the month/season for each problem
Line 73 Delete ‘-‘
Line 77-83 Is it a problem that the starlings eat invertebrates? Is it in scale so that it causes problems. Since I don’t know about this, it is difficult to understand that it is a problem.
Line 87 what ‘seeds’?
Line 93 ‘planting’ – sowing?
Line 94 ‘In Liguria (Italy) has been estimated a presence of 500,000 Starlings wintering for 80 days’ could be changed to ‘In Liguria (Italy), a presence of 500,000 Starlings wintering for 80 days was estimated’
Line 98 Delete ‘-‘
Line 102 Change ‘Starling also consumes’ to ‘Starlings also consume’
Line 107 Delete ‘-‘
Line 111 Starlings
Line 120 ‘…received damages…’ Is received the right word? Consider ‘Exposed to’ instead
Line 122 Delete ‘-‘
Line 128-138 Condiser moving this section to 1.1 or 1.2
Line 139-142 Difficult to read, consider rephrasing
Line 261 Capital Determination
Line 263 ..through (not throw)
Line 270-274 Difficult to understand, consider rephrasing
Line 279 Consider to change to ‘Finally, a heatmap showing the concentration of the trophic activity of the Starling according to the estimate of the average damage per hectare was created’
Line 317-318 7 roosts were identified with more than 20,000 starlings counted in at least one visit
Line 317-325 Make use of a full stop, please. I count 6 or 8 roosts > 20000 depending on where to to stop and start
Line 331 ‘a starling’ should be starlings. It seems to be a general problem throughout the paper. Please also decide if the first letter should be capital S or not.
Line 357 ‘…100 km…’ for foraging during the day? From the night roost?
Line 407-420 Change animals to birds
Line 414 Dormitories – could be ‘night roosts’ (several places in the text)
Line 441 I am not sure that ‘stolen’ is the right term
Line 513-515 I don’t understand this sentence
Reviewer 4 Report
The manuscript titled "Quantification of Starling population, estimation and mapping of the damage to olive crops in Apulia Region" addresses an interesting question on the damage caused by starlings to olive crops in a vast region of Italy. The manuscript is well-structured in general. However, the English style should be revised. Some parts of the manuscript are difficult to read because the ideas are expressed in a confusing way with really long sentences. Furthermore, I think that there is a lack of references in many parts of the text. Some data are provided without specifying the source. In general, I consider that the approach used by authors is interesting and it can be appreciated by potential readers (specially managers and local/regional governments). However, authors should clarify/improve some points. I have some comments that could improve the manuscript:
- Lack of references. This is a general comment that affects all the manuscript. But this are some examples:
- Line 38. Who said that Apulia is one of the most importan resting areas for many migratory species?
- Line 46-54. Where did you get all these information on the starling?
- Line 143-151. Cite those reports
- Line 171-184
- Line 357-358
- Line 401-406. Where did you get the info on the weight of olives? And the price?
- Line 426-431. Cite this preliminar investigation
- Line 97-110. Here, you said that starlings can feed on potential pests of the olives, is there any information/research about that? I think this could be something interesting to mention. They may be also beneficial.
- Line 172-173: “Techniques are taking into account….” I would say: Some techniques take into account….
- Line 175-177. Both sentences say the same
- Line 211 “individuation”. I do not understand this term, do you mean individualization? If this is the case, I think that you are not doing that in your study.
- Line 222. You should provide info on the sampling dates. For example in an Annex
- Line 261. “determination” change to “Determination”
- Line 270-272. This sentence should be re-structured.
- Line 287. Company? Do you mean farm?
- Line 331-337. This is an example of a extremely long sentence. Furthermore, it is hard to follow. Please re-structure it.
- Line 373-375. I think that these ideas can be expressed in a simpler way. The sentence is a little bit confusing.
- Line 388. It is “3.3”
- You used quintals as unit. Maybe it is common in Italy, but it would be a good idea to say once in the manuscript the equivalence to kilogrames. Many readers (like me) can not know this unit of measurement
- Line 415. “decade”? Maybe do you mean “fortnight”
- Which environmental variables did you use for the interpolation?
- Figure 5. Maps should include the location of the sampled areas.
Round 2
Reviewer 1 Report
From my point of view the applied methods are not scientific and presented results are questionable. Assessment of damage caused by starling is based rather on speculations not results of scientific investigation. As I understand it was a subjective opinion of farmers. How they assessed potential olive crop in particular year? Why they think that only starlings eat olives? Are there no others factors limiting olive crops, for example insects, fungus, other birds, mammals, weather….? Without well designed field experiment such assessment is impossible.
Such experiment is very easy: farmer should divide homogenous orchard in two plots and one of them should be protected against birds by nets, it is a common and efficient method applied in European cherry orchards. Finally, comparison of crop from two plots would give a real data about damage done by birds. Such experiment should be done in different locations and results extrapolated to the whole study area.
From my point of view the applied methods are not scientific and presented results are questionable. Assessment of damage caused by starling is based rather on speculations not results of scientific investigation. As I understand it was a subjective opinion of farmers. How they assessed potential olive crop in particular year? Why they think that only starlings eat olives? Are there no others factors limiting olive crops, for example insects, fungus, other birds, mammals, weather….? Without well designed field experiment such assessment is impossible.
Such experiment is very easy: farmer should divide homogenous orchard in two plots and one of them should be protected against birds by nets, it is a common and efficient method applied in European cherry orchards. Finally, comparison of crop from two plots would give a real data about damage done by birds. Such experiment should be done in different locations and results extrapolated to the whole study area.
From my point of view the applied methods are not scientific and presented results are questionable. Assessment of damage caused by starling is based rather on speculations not results of scientific investigation. As I understand it was a subjective opinion of farmers. How they assessed potential olive crop in particular year? Why they think that only starlings eat olives? Are there no others factors limiting olive crops, for example insects, fungus, other birds, mammals, weather….? Without well designed field experiment such assessment is impossible.
Such experiment is very easy: farmer should divide homogenous orchard in two plots and one of them should be protected against birds by nets, it is a common and efficient method applied in European cherry orchards. Finally, comparison of crop from two plots would give a real data about damage done by birds. Such experiment should be done in different locations and results extrapolated to the whole study area.
Reviewer 2 Report
Compared to the previous version of manuscript, the authors made many corrections.However, paper still needs some changes. The background analysed problem is not fully described.
Some sentences in results are reffering to the discussion. The results are to long and the discussion is to short.
Conclusion includes the contents of the previous chapters. Figure 3 is made without any edits and does not look like typical for this type of articles.
Reviewer 3 Report
Review Animals 2021, revised version
Quantification of starling population, estimation and mapping of the damage to olive crops in Apulia Region
By Francesco Bozzo, Simona Tarricone , Alessandro Petrontino, Prospero Cagnetta , Giacomo Maringelli , Giuseppe La Gioia , Vincenzo Fucilli , Marco Ragni
General comments:
The manuscript was improved considerably! However it is still necessary to have a thorough language editing (like most text by non-native English speakers). Hence, I only have minor comments.
Good luck in finalizing the paper
Minor comments:
Line 45 starlings (several places in the text, check each)
Line 47-48 Suggest to change to: The Starling (Sturnus vulgaris) is a gregarious medium-sized passerine with a short tail and long bill.
Line 48 Suggest to change to: In Apulia, the first breeding starlings were observed in Bari in approximately 1965
Line 50-51 I miss information about the population trend (over decades) during the period you study (November – February), i.e the winter population, if it is possible. I guess that the local breeding birds are only a minority of the birds). Are the local breeding birds sedentary?
Line 54 Change ‘that coincided’ to ’coinciding’
Line 60 starlings
Line 67 The Starling or the starling (s or S) – still not consistent
Line 82 ‘valuated’ could be changed to ‘estimated-?
Line 83 While A research program of Liguria Region had estimated
Line 85-86 No need to explain what a ton is; ‘with’ could be changed to ‘equivalent to’?
Line 126 region
Line 127 both ? ( I expected two things then)
Line 131 Reeds -> Reed beds
Line 132 this birds, it was -> these birds, we used
Line 143-145 Difficult to read, please rephrase
Line 151-152 Delete everything after the comma
Line 160 Is ‘particles’ the right word in ‘particles of 10 km UTM’?
Line 221 reeds were mentioned earlier in the text (move Phragmithes australis up)
Table 1 What is’1’ in ‘2016/2017 1’?
Line 245-248 How do you know the origin of the roosts?
Line 251 come mostly -> the majority arrive
Line 264 complained -> complained about
Line 277-278 Suggest to change to: the value of the loss was estimated at approximately 550,000 euros.
Line 279 800 thousand . be consistent (numerical above, text here)
Line 302-314 You will need to explain what the % is
Line 303 Explain in the fig text what % is
Line 305 Is >30% = red in the map?
Line 305-306 ? Should the sentence be part of the next sentence?
Line 326 7 most… In many journals this should be written as ‘seven most’ (numbers up to 10; several places in the text)
Line 348 found -> found that
Line 359 Suggest to change to; This means that during the olive growing season, c. a million…
Line 375-380 Difficult to read, please rephrase
Line 392 a -> as a
Line 402 Maybe just ‘To improve…’
Line 406 This to -> This will
Author Response
The manuscript was improved considerably! However it is still necessary to have a thorough language editing (like most text by non-native English speakers). Hence, I only have minor comments.
Good luck in finalizing the paper
Thank you very much, we have correct all the minor comments.
Minor comments:
Line 45 starlings (several places in the text, check each)
We corrected words into the text
Line 47-48 Suggest to change to: The Starling (Sturnus vulgaris) is a gregarious medium-sized passerine with a short tail and long bill.
We changed the sentence
Line 48 Suggest to change to: In Apulia, the first breeding starlings were observed in Bari in approximately 1965
We changed the sentence
Line 50-51 I miss information about the population trend (over decades) during the period you study (November – February), i.e the winter population, if it is possible.
We did not have information about the population trend in last decades; this was the first census of this population in Apulia region.
I guess that the local breeding birds are only a minority of the birds).
In fact
Are the local breeding birds sedentary?
Yes, they are
Line 54 Change ‘that coincided’ to ’coinciding’
We changed it
Line 60 starlings
done
Line 67 The Starling or the starling (s or S) – still not consistent
We have corrected words in the text
Line 82 ‘valuated’ could be changed to ‘estimated-?
We changed it
Line 83 While A research program of Liguria Region had estimated
We corrected the sentence
Line 85-86 No need to explain what a ton is; ‘with’ could be changed to ‘equivalent to’?
We corrected them
Line 126 region
done
Line 127 both ? ( I expected two things then)
We corrected the sentence
Line 131 Reeds -> Reed beds
done
Line 132 this birds, it was -> these birds, we used
We corrected the sentence
Line 143-145 Difficult to read, please rephrase
We rephrased it
Line 151-152 Delete everything after the comma
We delete all thing
Line 160 Is ‘particles’ the right word in ‘particles of 10 km UTM’?
We changed the word with greeds
Line 221 reeds were mentioned earlier in the text (move Phragmithes australis up)
We moved it
Table 1 What is’1’ in ‘2016/2017 1’?
Nothing, we deleted it
Line 245-248 How do you know the origin of the roosts?
This was our hypothesis because in the same day we monitoring roosts in the same area, so we know were starlings sleep in those days.
Line 251 come mostly -> the majority arrive
We changed it
Line 264 complained -> complained about
We corrected the sentence
Line 277-278 Suggest to change to: the value of the loss was estimated at approximately 550,000 euros.
We changed it
Line 279 800 thousand . be consistent (numerical above, text here)
We corrected them
Line 302-314 You will need to explain what the % is
We added the explanation into chapter 2.2
Line 303 Explain in the fig text what % is
We corrected the fig text
Line 305 Is >30% = red in the map?
yes
Line 305-306 ? Should the sentence be part of the next sentence?
We unified the sentences
Line 326 7 most… In many journals this should be written as ‘seven most’ (numbers up to 10; several places in the text)
We changed the number
Line 348 found -> found that
done
Line 359 Suggest to change to; This means that during the olive growing season, c. a million…
We changed it
Line 375-380 Difficult to read, please rephrase
We rephrased the paragraph
Line 392 a -> as a
done
Line 402 Maybe just ‘To improve…’
We corrected it
Line 406 This to -> This will
done